# Biofilm-Forming by Carbapenem Resistant Enterobacteriaceae May Contribute to the Blood Stream Infection

**DOI:** 10.3390/ijms20235954

**Published:** 2019-11-26

**Authors:** Kenichiro Yaita, Kenji Gotoh, Ryuichi Nakano, Jun Iwahashi, Yoshiro Sakai, Rie Horita, Hisakazu Yano, Hiroshi Watanabe

**Affiliations:** 1Chidoribashi General Hospital, Fukuoka 812-0044, Japan; 2Department of Infection Control and Prevention, Kurume University School of Medicine, Kurume 830-0011, Japan; 3Department of Microbiology and Infectious Diseases, Nara Medical University, Nara 634-0813, Japan; 4Department of Pharmacy, Kurume University Hospital, Kurume 830-0011, Japan; 5Department of Clinical Laboratory Medicine, Kurume University Hospital, Kurume 830-0011, Japan

**Keywords:** carbapenem-resistant Enterobacteriaceae, CRE, bloodstream infection, biofilm

## Abstract

Bloodstream infection (BSI) due to carbapenem-resistant Enterobacteriaceae (CRE) has a high mortality rate and is a serious threat worldwide. Ten CRE strains (eight *Enterobacter cloacae*, one *Klebsiella pneumoniae* and one *Citrobacter freundii*) were isolated from the blood of nine patients, a percentage of whom had been treated with indwelling devices. The steps taken to establish cause included minimum inhibitory concentration (MIC) tests, a pulsed-field gel electrophoresis (PFGE), biofilm study, a multiplex PCR for resistant genes of carbapenemases and extended-spectrum beta-lactamases (ESBLs), and plasmid incompatibility typing. All strains showed a tendency toward resistance to multiple antibiotics, including carbapenems. Frequently isolated genes of ESBLs and carbapenemases include *bla*_TEM-1_ (four strains), *bla*_SHV-12_ (four strains) and *bla*_IMP-1_ (six strains). A molecular analysis by PFGE was used to divide the *Xba*I-digested genomic DNAs of 10 CRE strains into eight patterns, and the analysis showed that three *E. cloacae* strains detected from two patients were either identical or closely related. The biofilm production of all CRE strains was examined using a microtiter biofilm assay, and biofilm growth in continuous flow chambers was observed via the use of a confocal laser scanning microscope. Our study indicates that biofilm formation on indwelling devices may pose a risk of BSI due to CRE.

## 1. Introduction

Hospital-acquired infection caused by carbapenem-resistant Enterobacteriaceae (CRE) is a serious threat for poor clinical outcomes, and CRE outbreaks have already been reported worldwide [1,2,3]. CRE is conferred either through the production of extended-spectrum β-lactamase (ESBL) and cephalosporinases (AmpC) combined with defective outer membrane (porin) permeability or via the acquisition of carbapenemase genes [4]. There are numerous different types of carbapenemase enzymes, such as the KPC, OXA-48-like, VIM, NDM, and IMP enzymes. The distributions of these enzymes differ by geographical location. For example the IMP enzyme is predominantly found in Japan [5]. The bloodstream infection (BSI) is a serious clinical problem due to multiple forms of the existence of antibiotic resistance and high mortality rates [6,7,8]. Microorganisms that cause BSI often produce biofilms. A biofilm is a structured community of bacterial cells that is enveloped in a self-produced, extra-polymeric matrix that is adherent to the surfaces of the materials [9]. Bacterial biofilms are recognized as important causes of a variety of human infections, including infections of prosthetic devices, endocarditis, dental caries, pneumonia in cystic fibrosis, and others [10].

The risk factors for BSI due to CRE include the use of broad-spectrum antibiotics, indwelling urethral catheterization, admission to the intensive care unit (ICU), respiratory infection, multiple comorbidities and repeated admissions to the hospital [7,8]. Those are mainly clinical risk factors, however, and the bacterial risk factors for BSI due to CRE remain unclear. 

Accordingly, we analyzed the association between bacterial characteristics such as antibiotic susceptibility, resistant mechanism, biofilm formation and clinical features.

## 2. Results

### 2.1. Bacterial Strains and Patient Characteristics

Between September 2014 and February 2016, there were 10 CRE strains (eight *Enterobacter cloacae*, one *Klebsiella pneumoniae* and one *Citrobacter freundii*) isolated from the blood samples of nine hospitalized patients. The clinical characteristics of the patients are summarized in Table 1. The mean age was 55.9 (range; 21–84), and six were male. Acute pyelonephritis was the most common diagnosis (four patients). Most (eight of nine) of the patients had indwelling devices such as a central venous catheter (four patients), a urethral catheter (four patients), or a urinary tract stent (three patients). Antibiotic-combination therapy had been administered to four patients, and other patients had been treated either meropenem, levofloxacin or cefepime. Two of the nine patients died. 

### 2.2. Antimicrobial Susceptibility, Resistance Genes and Transmission Frequencies

The characteristics of the isolated strains are summarized in Table 2, which shows the minimum inhibitory concentrations (MICs), harboring resistant genes, and transmission frequencies of our strains. *E. cloacae* was the most commonly detected pathogen ( eight strains in seven cases), followed by *K. pneumoniae* (one strain) and *C. freundii* (one strain). All strains showed a level of resistance to multiple antibiotics including carbapenems. Among the ESBL genes, *bla*_TEM-1_ and *bla*_SHV-12_ were dominant (four strains in each). Among the metallo-β-lactamase genes, only *bla*_IMP-1_ was detected in six strains. The incompatibility among the strains was unknown. Among six metallo-β-lactamase gene-harboring strains, the plasmids of two strains Patients 3 and 5) were transferable; however, the frequencies were low. 

### 2.3. Interpretation of Pulsed-Field Gel Electrophoresis (PFGE)

Molecular typing by pulsed-field gel electrophoresis (PFGE) showed that the *Xba*I-digested genomic DNAs of 10 CRE strains were divided into eight patterns (A–H) with no one pattern predominant (Figure 1). The PFGE patterns of two *E. cloacae* strains detected in Patients 2 and 7 were identical (F1) and those of two *E. cloacae* strains detected from Patient 7 were closely related.

### 2.4. Biofilm Study

The mean OD_595_ values of 10 CRE strains were all more than 2.0 in the microtiter biofilm assay, which indicated a high activity of biofilm formation (Figure 2). In order to reveal the production of biofilms under flow pressure, the CRE strain (Case No.1) was studied in a continuous flow chamber. A confocal microscopic analysis indicated that the CRE strain produced biofilm. The upstream biofilm in the chamber had a thickness of approximately 20–22 μm with a relatively smooth surface, but the downstream biofilm was approximately 30–35 μm thick with an irregular surface (Figure 3).

## 3. Discussion

In this study, we characterized the CRE isolated from the blood samples of nine hospitalized patients in the Kurume University Hospital of Japan. Six of the 10 CRE isolates produced IMP-1, but the remaining four isolates had produced no carbapenemases. Despite the fact that KPC, OXA-48, and NDM are globally disseminated, they are rarely found in Japan, where IMP-1 and IMP-6 are the exclusively predominant carbapenemases [11,12]. Our study results revealed that IMP-1-producers were dominant in this hospital. Three of the *E. cloacae* isolates (from Patients 2 and 7) had almost the same PFGE patterns and were estimated to be nearly identical isolates. The other three isolates, including *K. pneumoniae*, *E. cloacae*, and *C. freundii*, were also harboring *bla*_IMP-1_ and the coding plasmid could transfer with frequencies of 10^−6^–10^−7^. The *bla*_IMP-1_ coding plasmid was the suspected transmitting agent among the species. Further experimentation is required to clarify this mechanism. All the non-carbapenemase producing CRE were *E. cloacae* isolates, and they were estimated to be resistant to carbapenems due to the production of AmpC combined with a defective outer membrane (porin) permeability. Previous reports have established that all patients in Japan with *E. cloacae* harboring *bla*_IMP-1_ and *bla*_IMP-11_ had indwelling devices at the time of isolation, which, along with exposure to cephalosporin and a history of invasive procedures, were associated with the isolation of IMP-producing *E. cloacae* [13]. 

Most (eight of nine) of the patients in this study had indwelling devices such as a central venous catheter, a urethral catheter, or a urinary tract stent, a fact which is consistent with previous reports [7,13]. There remains scant information regarding the relationship between BSI and biofilm formation by CRE. However, in our study, all 10 CRE isolates from blood samples formed biofilm according to a microtiter biofilm assay and a biofilm study in a continuous flow chamber, although most of the PFGE patterns differed. Biofilm formation is known to be related to BSI in organisms such as *Candida, Staphylococcus* and *Corynebacterium* species. Also, formed biofilm is known to harbor bacteria that are more resistant to antibiotic therapy than are plank-tonic organisms [14,15,16,17]. Considering our results and facts, it is thought that these strains with a high biofilm production performance escaped the load of antibiotics, settled in each individual, and caused infectious diseases. A further problem is that these strains can potentially acquire resistance to various antimicrobial agents because of their high ability to produce biofilms. This is considered a risk for the emergence of multidrug resistant CRE. 

A limitation of our study was that the analysis was of a small number of isolates only from blood cultures. Further studies should include isolates from other cultures such as urine or sputum and further elucidation of the various drug resistance mechanisms of CRE.

## 4. Materials and Methods 

### 4.1. Ethical Approval

All studies described herein were approved by the Human Ethics Review Boards of Kurume University on 12 Feb 2016 (Research No. 15239).

### 4.2. Bacterial Strains and Patients

Between September 2014 and February 2016, there were 10 CRE strains (8 *Enterobacter cloacae*, one *K. pneumoniae*, and one *Citrobacter freundii*) that were isolated from blood samples of 9 hospitalized patients in Kurume University Hospital, which is a 1025-bed tertiary care medical center. We retrospectively checked the electronic medical charts of participants and classified their factors as follows: age, sex, primary infection due to CRE, underlying disease, indwelling devices, antibiotic therapies, and outcomes. 

### 4.3. MIC

The MICs of cefepime, cefmetazole, imipenem, meropenem, piperacillin/tazobactam, amikacin and levofloxacin were determined in reference to WalkAway96plus (Beckman Coulter, Inc., Brea, CA, USA) via the broth microdilution method in accordance with the guidelines of the Clinical and Laboratory Standards Institute [14]. The criteria of the CRE were based on laboratory findings of Japanese criteria (http://www.nih.go.jp/niid/images/iasr/35/418/de4181.pdf) as follows:

The MIC for meropenem was ≥2 μg/mL, the MIC for imipenem was ≥2 μg/mL, and the MIC for cefmetazole was ≥64 μg/mL.

### 4.4. Molecular Analysis of Antimicrobial Resistance Genes

CRE strains were subjected to multiplex PCR for resistant genes, carbapenemases (*bla*_IMP,_
*bla*_VIM,_
*bla*_OXA-48,_
*bla*_NDM_, and *bla*_KPC_) and ESBLs (*bla*_TEM,_
*bla*_SHV,_
*bla*_OXA,_
*bla*_CTX-M_), as found by using methods described previously [18,19,20]. The primers for the sequencing of *bla*_IMP_ were 5′-TGA GCA AGT TAT CTG TAT TC-3′ (forward primer) and 5′- TTA GTT GCT TGG TTT TGA TG-3′ (reverse primer) [21]. The primers for the sequencing of ESBLs were 5′-CAT TTC CGT GTC GCC CTT ATT C-3′ (forward) and 5′-CGT TCA TCC ATA GTT GCC TGA C-3′ (reverse) for *bla*_TEM_ and 5′-AGC CGC TTG AGC AAA TTA AAC-3′ (forward) and 5′-ATC CCG CAG ATA AAT CAC CAC-3′ (reverse) for *bla*_SHV_ [18,19]. In the *bla*_CTX-M_ group, only *bla*_CTX-M-9_ could be detected, and then we used the primers 5′-ATG GTG ACA AAG AGA GTG CAA CGG-3′ (forward) and 5′-TCA CAG CCC TTC GGC GAT GAT TCT-3′ (reverse) [22]. The plasmid incompatibility typing of our strains was performed via PCR by using primers of the 18 major incompatibility groups (FIA, FIB, FIC, HI2, I1-Ic, L/M, N, P, W, T, A/C, K, B/O, X, Y, F, FIIAm including the IncHI1 type) among Enterobacteriaceae [23]. 

### 4.5. Conjugal Transfer of Plasmid

In vitro conjugation experiments were performed using the broth method, as previously described [24]. Mid-log phase LB-broth cultures of donor strains and the recipient strain *Escherichia coli* J53 were mixed at a volume ratio of 1:10. This mating mixture was incubated for 2 h at 35 °C. The transconjugants were selected on an LB plate that included sodium azide (100 μg/mL) and cefpodoxime (8 μg/mL). The transfer frequencies were expressed as transconjugants per recipient cell.

### 4.6. PFGE

The PFGE of 10 CRE strains was performed, as described previously [25]. The DNA was digested with *Xba*I (Takara Shuzo Co., Shiga, Japan). CHEF Mapper pulsed field electrophoresis systems (Bio-Rad Life Science Group, Hercules, CA, USA) were used with a potential of 6 V/cm, switch times of 0.47 and 63 s, and run-times of 20 h and 18 min. After staining with ethidium bromide, the PFGE patterns were interpreted based on the criteria described by Tenover et al. [26]. 

### 4.7. Microtiter Biofilm Assay

CRE biofilms were investigated via a modified microtiter biofilm assay [27]. Bacteria cells were suspended in a Mueller Hinton broth at approximately 0.1 OD_490_, and 200 μL aliquots were inoculated into the wells of 96-well plates with a broth medium as negative control. Biofilm formation was confirmed after 48 h of culture. Finally, biofilms were stained with 1% freshly adjusted crystal violet (MERCK Millipore, Meguro, Tokyo, Japan) at room temperature for 15 min and vigorously washed three times with water. Following extraction with 230 μL of 95% ethanol, the biofilms were measured at OD_595_ with a micro plate reader. All the tests were performed in triplicate. 

### 4.8. Biofilm Growth in Continuous Flow Chambers

The CRE strain (Case No.1) was grown in continuous flow chambers for in vitro biofilm growth. The chambers were constructed with the same dimensions and materials as previously described [28], but the outflow fitting material was changed from copper to plastic. Glass coverslips (22 × 50 mm) were placed on the chambers and stabilized with silicone. The chambers with coverslips and the influent and effluent tubing all were sterilized via autoclaving. The CRE strain was grown to mid-log phase in a Mueller Hinton broth. The chamber with clamped effluent tubing on a flat surface was inoculated with 6 mL of cultures diluted to OD_595_ = 0.25. The chamber was incubated at 37 °C for 2 h to allow bacteria to adhere to the coverslip. The influent tubing was aseptically connected to the flask containing the Roswell Park Memorial Institute(RPMI )media, diluted 1:4 in Phosphate buffered salts (PBS), and then to the chamber via a 23-gauge needle (1 inch), which was aseptically inserted through the inlet stopper. Both the inflow and outflow tubing were fed through the same pump to maintain a constant volume in the chamber. The chamber, on a flat surface, was incubated at 37 °C for 48 h at a flow rate of 100 μL/min. When indicated, the biofilm was stained using a LIVE/DEAD BacLight Bacterial Viability Kit according to the manufacturer’s instructions (Life Technologies) prior to fixation. The sample was viewed by confocal laser-scanning microscopy using an Olympus FLUOVIEW FV3000 laser scanning confocal microscope (Olympus corporation, Shinjyuku, Tokyo, Japan) at a magnification of 20×. 

## 5. Conclusions

In conclusion, our study indicates that indwelling devices are a potential risk for BSI due to CRE, and it implicates that CRE adhere to indwelling devices and produce biofilm. Therefore, we may consider the removal of indwelling devices related to biofilm formation in the treatment of BSI due to CRE.

## Figures and Tables

**Figure 1 ijms-20-05954-f001:**
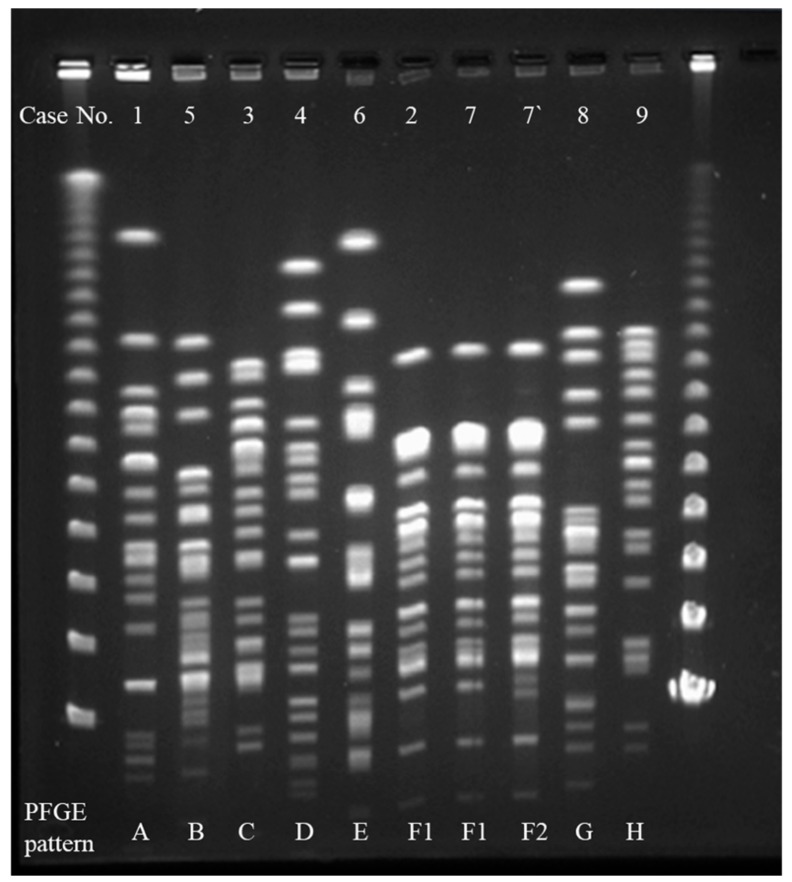
Pulsed-field gel electrophoresis (PFGE) patterns of *Xba*I-digested DNA from 10 carbapenem-resistant. Enterobacteriaceae isolates. Ten Enterobacteriaceae (CRE) strains had eight PFGE patterns (A–H) with no predominant pattern. The PFGE patterns of two *E. cloacae* strains detected from Patients 2 and 7 were identical (F1) and those of two *E. cloacae* strains detected from Patients 7 and 7′ were closely related (F1 and F2, respectively).

**Figure 2 ijms-20-05954-f002:**
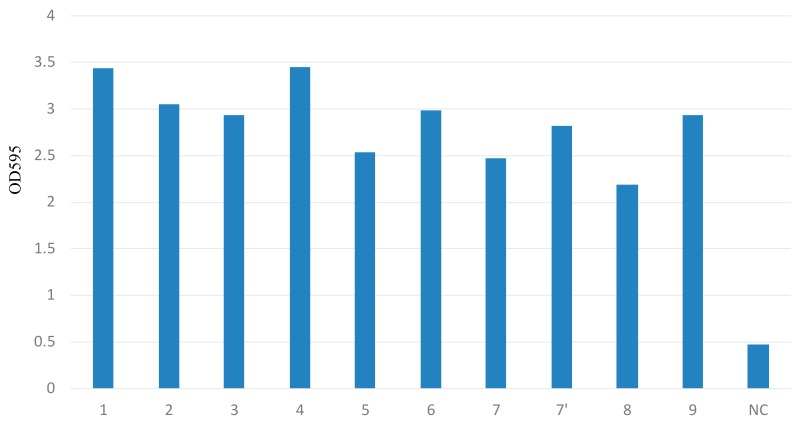
Results of the microtiter biofilm assay. The mean OD_595_ values of 10 CRE strains were all more than 2.0, and biofilm formations were confirmed.

**Figure 3 ijms-20-05954-f003:**
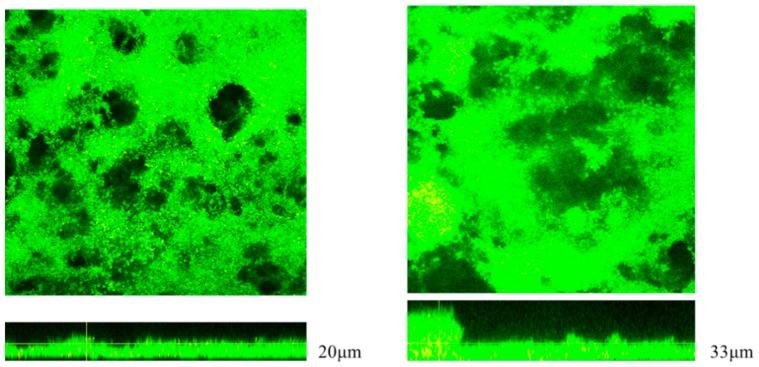
Results of the drip-flow biofilm assay. A confocal microscopic analysis indicated the upstream biofilm in the chamber had a thickness of approximately 20–22 μm with a smooth surface. The biofilm thickness in the downstream portion of the chamber, however, was approximately 30–35 μm.

**Table 1 ijms-20-05954-t001:** Patient characteristics.

Case No.	The Month of Detection CRE	Age, Sex	Primary Infection	Underlying Disease	Indwelling Devices	Antibiotic Therapy	Outcome
1	2014.11	40, Male	Acute pyelonephritis	Chronic kidney disease, Hemodialysis	Urinary tract stent	Meropenem+Ciprofloxacin	Survive
2	2015.2	62, Male	Febrile neutroenia	Diffuse large B-cell lymphoma, Bone marrow transplantation	Central venous catheter	Meropenem	Death
3	2015.2	21, Female	Catheter-related blood stream infection	Mixed connective tissue disease, Acute kidney injury, Hemodialysis	Central venous catheter, Urethral catheter	Piperacillin/tazobactam+Tigecycline	Death
4	2015.7	71, Male	Acute pyelonephritis	Renal cell carcinoma	Urinary tract stent, Urethral catheter	Tigecycline+Amikacin	Survive
5	2015.8	79, Female	Obstructive cholangitis	Papilla cancer, Diabetes mellitus	Retrograde transhepatic biliary drainage tube	Levofloxacin	Survive
6	2015.8	56, Male	Acute pyelonephritis	Urinary tract stone, Diabetes mellitus	Urinary tract stent, Urethral catheter	Meropenem	Survive
7	2015.9	59, Female	Vertebral osteomyelitis	Diffuse large B-cell lymphoma, Lung cancer	Central venous catheter	Tigecycline+Amikacin	Survive
8	2015.11	84, Male	Acute pyelonephritis	Benign prostatic hyperplasia	None	Cefepime	Survive
9	2015.12	31, Male	Primary bacteremia	Severe burn, acute respiratory distress syndrome, Diabetes mellitus	Central venous catheter, Urethral catheter	Levofloxacin	Survive

**Table 2 ijms-20-05954-t002:** Minimum inhibitory concentrations, resistant genes, and transmission frequencies of our strains.

Case No.	Species	CFPM	CMZ	IPM	MEPM	P/T	AMK	LVFX	Resistant Genes	Transfer Frequency
1	*Klebsiella pneumoniae*	>16	>32	2	>2	64	<4	<0.5	*bla* _IMP-1_ *, bla* _TEM-1_ *, bla* _SHV-12_	not transferred
2	*Enterobacter cloacae*	16	>32	<1	2	64	<4	2	*bla* _IMP-1_ *, bla* _SHV-12_	not transferred
3	*Enterobacter cloacae*	16	>32	<1	>2	<16	<4	>4	*bla* _IMP-1_	4.0 × 10^−7^
4	*Enterobacter cloacae*	>16	>32	2	2	>64	16	>4	*bla* _TEM-1_ *, bla* _SHV-12_	not determined
5	*Citrobacter freundii*	<2	>32	2	>2	<16	<4	<0.5	*bla* _IMP-1_	1.8 × 10^−6^
6	*Enterobacter cloacae*	>16	>32	2	<1	64	<4	>4	bla_CTX-M-27_	not determined
7	*Enterobacter cloacae*	16	>32	>2	>2	>64	<4	>4	*bla* _IMP-1_ *, bla* _TEM-1_ *, bla* _SHV-12_	not transferred
7′	*Enterobacter cloacae*	>16	>32	>2	>2	64	<4	>4	*bla* _IMP-1_ *, bla* _TEM-1_	not transferred
8	*Enterobacter cloacae*	<2	>32	2	<1	<16	<4	<0.5	not detected	not determined
9	*Enterobacter cloacae*	<2	>32	2	<1	<16	<4	<0.5	not detected	not determined

CFPM = cefepime, CMZ = cefmetazole, IPM = imipenem, MEPM = meropenem, P/T = piperacillin/tazobactam, AMK = amikacin, LVFX = levofloxacin.

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
