# Peer review of "Biofilm-Forming by Carbapenem Resistant Enterobacteriaceae May Contribute to the Blood Stream Infection"

_ijms, 2019, doi:10.3390/ijms20235954_

Round 1

Reviewer 1 Report

The manuscript described a small-scale characterization of 10 carbapenem-resistant Enterobacteriaceae isolated from blood. The properties being investigated were antibiotic susceptibility to carbapenem, cephalosporin, aminoglycoside and fluoroquinolone. Other properties studied were genes associated with antibiotic resistance, biofilm formation & clonal relatedness. Although the number of samples are small, the detailed analyses should yield informative background of the strains. However, the manuscript has not provided an in-depth discussion of the results obtained. The relationship between various tests was not sufficiently discussed. Some results presented in the manuscript (e.g. transconjugation efficiency) had no description on the methodology. There is no mention on the used of control in the biofilm assay.

Reviewer 2 Report

General comments

This short communication has significant scientific value as it meets their impact on regional as well as global public health. The article focuses on the spreading of CRE, and their resistance determinants originated from the hospital in Japan. This research is of high interest and may help in preventive measures.

However, several clarifications and corrections are needed in the manuscript before the publication. Also, I would recommend the authors for English correction to enhance the flow and readability.

Title

Please rephrase the title.

Abstract

Restructure line 20-21

The authors may consider as following.

“Blood stream infection (BSI) due to Carbapenem-resistant Enterobacteriaceae (CRE) is a serious threat worldwide…… “

Line no 34, put (MIC).

Line no 27 -28. The author has mentioned ESBL and carbapenemase. Please specify the gene names appropriately. Put the names of blaTEM-1 and blaSHV-12 first and then blaIMP-1.

Line no 33- 34. Rephrase the sentence for clarity.

Introduction

Line 41 – 45. It is not clear.  Please add the mechanisms substantiated by relevant literature citations. Moreover, the authors did not differentiate well the mechanisms and types of CRE.

Line no 45- 46. Please rephrase the sentence.

On the whole, the introduction lacks clear information and sequence. I would suggest the authors add some points concerning Biofilms.  

Results

2.1 Bacterial strains and patient characteristics

Line no 60 -61. Please restructure for the clarity.

Table 1. The Year could be included.

I would recommend the author to put the section title as patient characteristics. No information about bacterial strains in this section.

Antimicrobial susceptibility, resistance genes and transmission frequencies

Line no 64… The characteristics of our strains. Please correct to the characteristics of “isolated strains”: harboring of resistant genes to “harboring resistant genes”.

Did authors do DNA sequence analysis for identified resistance genes?

Discussion

Line no 100. It seems repetition. ” although the mechanisms of drug resistance in CRE were numerous”. What are they and could be better described.

Line no 100 -103. Please restructure the sentence.

Line no 106 -110. It could be moved to the introduction.

Please give a detail discussion substantiated with previous findings to support the present results.

Materials and methods

Biofilm growth in continuous flow chambers

Please correct - The CRE strain (No.1) to The CRE strain (Case No.1)

Conclusion

It could be improved and should reflect the major outcomes of this study.   

Round 2

Reviewer 1 Report

No further.

Reviewer 2 Report

The manuscript has now improved much better than the first version. Authors have answered satisfactorily to all of my comments. Thanks.